# Margin-Based Few-Shot Class-Incremental Learning with Class-Level Overfitting Mitigation

**Yixiong Zou[1], Shanghang Zhang[2], Yuhua Li[1] and Ruixuan Li[1]***

[1]School of Computer Science and Technology, Huazhong University of Science and Technology
[2]School of Computer Science, Peking University
[1]{yixiongz, idcliyuhua, rxli}@hust.edu.cn, [2]shanghang@pku.edu.cn

## Abstract

Few-shot class-incremental learning (FSCIL) is designed to incrementally recognize novel classes with only few training samples after the (pre-)training on base classes with sufficient samples, which focuses on both base-class performance and novel-class generalization. A well known modification to the base-class training is to apply a margin to the base-class classification. However, a dilemma exists that we can hardly achieve both good base-class performance and novel-class generalization simultaneously by applying the margin during the base-class training, which is still under explored. In this paper, we study the cause of such dilemma for FSCIL. We first interpret this dilemma as a class-level overfitting (CO) problem from the aspect of pattern learning, and then find its cause lies in the easily-satisfied constraint of learning margin-based patterns. Based on the analysis, we propose a novel margin-based FSCIL method to mitigate the CO problem by providing the pattern learning process with extra constraint from the margin-based patterns themselves. Extensive experiments on CIFAR100, Caltech-USCD Birds-200-2011 (CUB200), and *mini*ImageNet demonstrate that the proposed method effectively mitigates the CO problem and achieves state-of-the-art performance.

## 1  Introduction

With the development of deep learning, deep neural networks gradually demonstrate superior performance on the recognition of pre-defined classes with large amount of training data [19, 10]. However, the model's generalization capability on the downstream novel classes is much less explored and still needs to be improved [13, 9]. To deal with this problem, the few-shot class-incremental learning (FSCIL) task [11, 18, 2, 22, 28, 30] comes into sight. FSCIL first (pre-)trains a model on a set of pre-defined classes (base classes), and then generalizes the model to the incremental novel classes with only few training samples, simulating human's ability of continually learning novel concepts with only few examples, and emphasizing both the performance on the pre-defined base classes and the generalization on the downstream novel classes.

However, a dilemma is recently revealed [13, 5, 6, 7] that better loss functions, which lead to higher performance on the pre-training data, could lead to worse generalization on the downstream tasks. As introduced by [16] and depicted in Fig. 1, similar phenomenon also exists in the FSCIL task that a positive classification margin [16, 25, 8, 21] applied to the classification of the base-class (pre-)training could lead to higher base-class performance but lower novel-class performance, while a negative margin could result in lower base-class performance but increase the novel-class performance. Although this dilemma widely exists in the tasks involving novel-class generalization such as few-shot learning (FSL) and FSCIL, only few works [16] tried to explore its cause, and can hardly be used to handle it. Due to space limitation, we will provide extended related works in the appendix.

---

*Corresponding author.

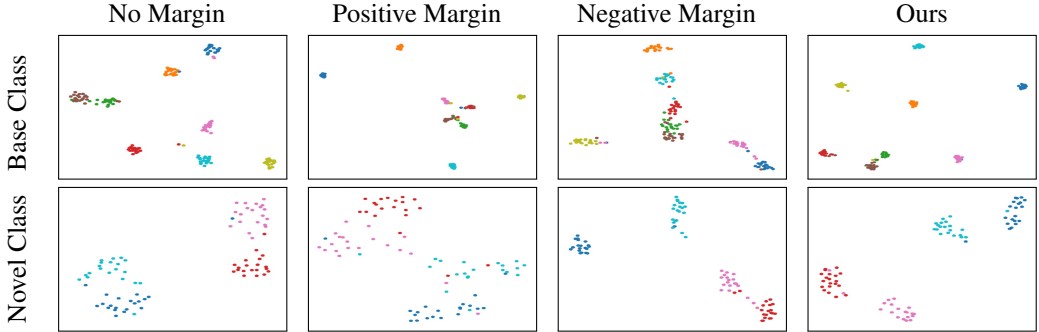

Figure 1: A dilemma exists between base-class performance and downstream novel-class generalization. By applying positive margins, base-class features are better separated which indicates better base-class performance, but the novel-class features are confused which indicates lower novel-class generalization. In contrast, by applying negative margins, base-class features are confused but the novel-class features are better separated. In this paper, we study the cause of such dilemma for the few-shot class-incremental learning task, and propose a method to mitigate such dilemma to better separate both base and novel classes.

In this paper, we study the cause of the dilemma in the margin-based classification for the FSCIL problem from the the aspect of pattern learning. We find this dilemma can be understood as a class-level overfitting (CO) problem, which can be interpreted by the fitness of the learned patterns to each base class. The fitness determines how much the learned patterns are specific to some base classes or shared among classes, making the learned patterns either discriminative (tend to overfit base classes) or transferable (tend to underfit base classes) and causes the dilemma. Based on the interpretation, we discover the cause of the dilemma lies in the easily-satisfied constraint of learning shared or class-specific patterns. Therefore, we further design a novel FSCIL method to mitigate the dilemma of CO by providing the pattern learning process with extra constraint from margin-based patterns themselves, improving performance on both base and novel classes as shown in Fig. 1, and achieving state-of-the-art performance in terms of the all-class accuracy. Our contributions are:

- We interpret the dilemma of the margin-based classification as a class-level overfitting problem from the aspect of pattern learning.

- We find the cause of the class-level overfitting problem lies in the easily-satisfied constraint of learning shared or class-specific patterns.

- We propose a novel FSCIL method to mitigate the class-level overfitting problem based on the interpretation and analysis of the cause.

- Extensive experiments on three public datasets verify the rationale of the model design, and show that we can achieve state-of-the-art performance.

## 2 Interpreting the Dilemma of Few-Shot Class-Incremental Learning

In this section, we first describe the Few-Shot Class-Incremental Learning (FSCIL) task and the baseline model, and then conduct experiments to analyze the dilemma.

### 2.1 Task and Baseline Description

The FSCIL task aims to incrementally recognize novel classes with only few training samples. Basically, the model is first (pre-)trained on a set of base classes with sufficient training samples (a.k.a. base session), then confronted with novel classes with limited training samples (a.k.a. incremental session), and finally required to recognize test samples from all encountered classes.

Specifically, given the base session dataset $D^0 = \{(x_i, y_i)\}_{i=1}^{n_0}$ with the label space $Y_0$, the model is trained to recognize all $|Y_0|$ classes from $Y_0$ by minimizing the loss

$$\sum_{(x_i, y_i) \in D^0} L(\phi(x_i), y_i), \tag{1}$$

where $L(\cdot, \cdot)$ is typically a cross-entropy loss, $\phi(\cdot)$ is the predictor which is composed of a backbone network $f(\cdot)$ for feature extraction and a linear classifier, represented as $\phi(x) = W^\top f(x)$ where $\phi(x) \in R^{N_0 \times 1}$, $W \in R^{d \times N_0}$ and $f(x) \in R^{d \times 1}$. Typically, $f(x)$ and the $W$ are $L_2$ normalized [28].

When the $k$th incremental session comes, the model needs to learn from its training data $D^k = \{(x_i, y_i)\}_{i=1}^{n_k}$. The weight of the classifier will be extended to represent the novel label space $Y_k$ imported by this session, represented as $W = \{w_1^0, w_2^0, ..., w_{|Y_0|}^0\} \cup ... \cup \{w_1^k, ..., w_{|Y_k|}^k\}$ where $w_j^k$ denotes the weight of the classifier corresponding to the $j$th class of the $k$th session.

A strong baseline [28] is to freeze model's parameters to avoid the catastrophic forgetting brought by the finetuning on novel-classes. For the incremental sessions (i.e., $k > 0$), the average of the features extracted from the training data will be used as the classifier's weight [28] (a.k.a. prototype) as $w_j^k = \frac{1}{n_k^j} \sum_{i=1}^{n_k^j} f(x_i)$, where $n_k^j$ denotes the number of training samples in the class $j$ for the session $k$. As this baseline focuses on the base-class training, in this paper, the term *training*, if not otherwise stated, refers to the base-class training. Finally, the performance of the $k$th session will be obtained by classifying the test samples from all $\sum_{i=0}^{k} |Y_i|$ encountered classes.

## 2.2 Margin-Based Classification

A well known modification to base-class training loss (Eq. 1) is to integrate a margin [16, 8, 25] as

$$L(x_i, y_i) = -log \frac{e^{\tau(w_{y_i} f(x_i) - m)}}{e^{\tau(w_{y_i} f(x_i) - m)} + \sum_{j \neq y_i} e^{\tau w_j f(x_i)}}, \tag{2}$$

where $w_{y_i}$ refers to the classifier weight for class $y_i$, $\tau$ is typically set to 16.0 and $m$ is the margin.

As analyzed in [16], empirically a dilemma exists that a positive margin could improve the base-class performance but harm the novel-class generalization, and reversely, a negative margin could contribute to the novel-class performance but decrease that of the base classes as shown in Fig. 1. Similar phenomenon has been observed in other works such as [13] that a better loss function for the pre-training task could harm the generalization on downstream tasks.

## 2.3 Interpretation of Class-Level Overfitting from Pattern Learning View

Experiments are conducted on the CIFAR100 [15] dataset and reported in Fig. 2. CIFAR100 contains 100 classes in all. As split by [22], 60 classes are chosen as base classes, and the remaining 40 classes (with 5 training samples in each class) are chosen as novel classes[2]. Experiments are conducted on the last incremental session, where all 100 classes are involved.

From Fig. 2 (left), we can see that as the margin increases, the base-class accuracy increases while the novel-class accuracy decreases, which is consistent with [16] and validates the dilemma exists. Compared with the well-known overfitting between the training and testing data, such dilemma, although all validated on testing data, is more like the overfitting to base classes instead of samples. Therefore, we term it as **class-level overfitting (CO)**. Additionally, the balance is reached when no margin is added, i.e., FSCIL cannot be improved by simply applying the margin. For such dilemma, [16] gave an explanation by the degraded mapping from novel to base classes. However, it could hardly be used to develop methods for handling such dilemma. In this section, we go a step further to explain this phenomenon from the aspect of pattern learning for developing methods to handle it.

A pattern denotes a part of information that the model extracts from the input, which is a finer-grained level of analyzing the model's behavior. As studied in the interpretability of deep nets [29, 1], each channel in the feature extracted by deep networks could correspond to a certain pattern of the input[2], which can be viewed as to compose the base and novel classes [31]. Therefore, we conduct experiments on feature channels to study the patterns learned by applying different margins.

### 2.3.1 Class-Level Overfitting Interpreted by Pattern Fitness to Base Classes

**Pattern's fitness to each base class.** We first evaluate the sparsity of the base-class patterns, which is measured by the $L_1$ norm of each feature vector. As the extracted features are $L_2$ normalized, the smaller the $L_1$ norm is, for each feature, the sparser the patterns with high activation are. Results are plotted in Fig. 2 (mid), where we can see a consistent decrease in $L_1$ when the margin increases, which means the model needs less activated patterns (channels) to represent each base class. As the number of activated patterns decreases, the effectiveness of each activated pattern must increase to account for the performance increase in the base-class pre-training in Fig. 2 (left). Therefore, we hold that as the margin increases, the patterns learned by the model could fit each base class better.

---

[2]Please refer to the appendix for details.

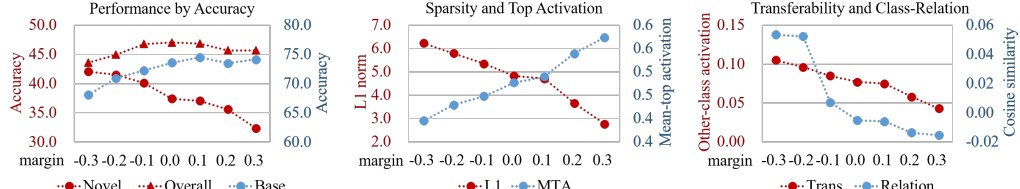

Figure 2: **Left**: Class-level overfitting exists between base and novel classes, and simply applying margins to the training can not help the overall performance. **Mid**: Pattern fits base classes more as the margin increases, making it more discriminative but less transferable. **Right**: Transferability of patterns decreases as the margin increases, pushing classes away from each other.

**Pattern fitness measured by the template-matching score.** To further verify the fitness increase, we view each pattern as a semantic template and measure its matching score to each base class. As analyzed in [29] and [1], each pattern can be understood as a template [3] for the model to match the input (so that each class would has its own set of templates for recognition), and the activation can be viewed as the matching score. Therefore, we could know how much all patterns fit (match) each class by finding the most important patterns for each class and compare their activation. As analyzed in [29] and [31], patterns (channels) with higher weights in the classification layer are more important, and the most important ones dominate the model decisions. Therefore, given an input, we select its most important patterns by the top classification weights of its ground-truth class, and record the average activation on these patterns. The Mean value of such Top Activation across all samples is denoted as MTA in Fig. 2 (mid). As can be seen, as the margin increases, MTA increases consistently, which further verifies patterns' increase in fitting each base class.

**Better pattern fitness, worse pattern transferability.** As each pattern could fit a corresponding base class better, its discriminability increases accordingly, but could it be transferred across classes? To answer it, we test the transferability of patterns. Since classes are related (e.g., cat and tiger), transferable patterns activated in one class could also be activated in other classes (e.g., felid patterns). Therefore, we first find important patterns for each base class by the classification weights, then record activation of these patterns on **other** classes, and measure the transferability of patterns by the mean value of such other-class-activation. The results are plotted in Fig. 2 (right). As can be seen, the transferability consistently decreases when the margin increases. Combine this result with Fig. 2 (mid), we hold that patterns tend to be less transferable when they fit each base class better.

**Discussion.** The fitness also reflects the how much the given pattern is specific to a base class. Imagine the extreme situation where each base-class only needs one pattern for representation, the fitness would reach its upper bound to make such pattern thoroughly specific to the corresponding class. Therefore, we interpret that the higher the margin is, the more specific (overfitting) the patterns are to each base class, which makes patterns more discriminative but less transferable. Meanwhile, the lower the margin is, the more the patterns could be shared between classes (underfitting), making patterns more transferable but less discriminative. The CO dilemma lies in that patterns can hardly be both class-specific and shared among classes by simply applying the classification margin.

### 2.3.2 Inherent Class Relations Lead to the Change in Pattern's Base-Class Fitness

**Pattern's fitness negatively influences class relations.** In Fig. 2 (right), we also plot the class relations w.r.t. the margins. The class relations are measured by the average of cosine similarities between every two classes' prototypes. As can be seen, the relation drops as the margin grows, in consistent with the trend of the patterns' transferability. This is rationale because the if two prototypes share some patterns, the activation of the corresponding channels will be similar, making the cosine similarity larger. As the transferability of patterns is negatively related to pattern's base-class fitness, we hold that the class relations are also negatively related to the base-class fitness.

**Inherent class relations influence pattern's fitness.** The margin applied to the classification directly modifies the decision boundary between every two classes, and the decision boundary is related to the relationship between every two classes. Therefore, we study how the class relation influences the pattern's fitness to base classes. Specifically, given 60 base classes, for the model trained without margins, we first calculate the cosine similarity between every two different classes, which gives $60 \times (60 - 1) / 2 = 1,750$ relations denoted as $R_0$, which represents the **inherent relations** between all classes. Similarly, we calculate 1,750 relations for the model trained with positive and negative margins respectively, denoted as $R_{pos}$ and $R_{neg}$. Then we calculate $D_{pos} = R_{pos} - R_0$

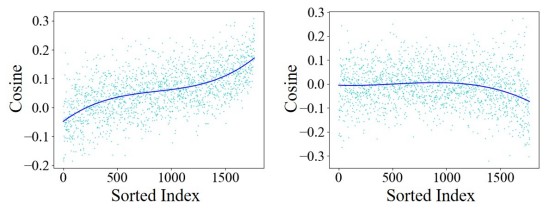
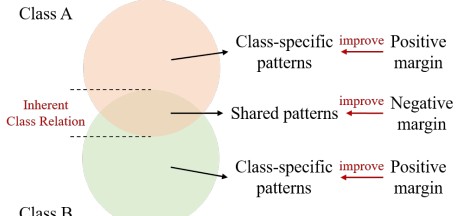

Figure 3: The change of class relations sorted by the inherent class relations. Left: negative margin. Right: positive margin.

Figure 4: Intuitive interpretation of the pattern learning and inherent class relation.

and $D_{neg} = R_{neg} - R_0$. Finally, we sort $R_0$ from small to large, and use the sorted index to arrange $D_{neg}$ and $D_{pos}$. As plotted in Fig. 3, the blue dots denote $D_{neg}$ (left) or $D_{pos}$ (right), and the blue curve are dots fitted by multinomial functions. We can see that the change in class relations is positively related to the inherent class relations for the negative margins, while is negatively related to the inherent class relations for the positive margins, especially for index larger than 1,000.

Since class relations are negatively related to pattern's fitness to base classes, the more the class relation increases, the less the patterns could fit the corresponding base class. Therefore, the results can be understood that the more inherently similar two classes are, by applying a negative margin, the less the patterns could fit the given base class, i.e., the more the patterns are shared by given classes, making these classes' representations more similar; by applying a positive margin, the more the patterns could fit the given base classes, making these classes' representations more dissimilar.

**Conclusion and Discussion.** Therefore, as shown in Fig. 4, we interpret the pattern learning process as follows. Given a negative margin, the decision boundary between two classes are confused, making more samples fall into the overlapping region between two classes. This makes it possible to learn patterns shared by these two classes and hinder the pattern from fitting the given base class, and makes patterns more transferable but less discriminative. The more similar inherently two classes are (i.e., larger overlapping region in Fig. 4), the more shared patterns can be learned (e.g., more patterns could be shared between cats and tigers than cats and air-planes), therefore making these classes' representations more similar. On the contrary, given a positive margin, the decision boundary between two classes should be well separated, pushing the model to learn patterns fitting (i.e., specific to) each class, which are more discriminative but less transferable. The more similar inherently two classes are, the harder the learning is and the larger the training loss will be, therefore making the patterns fit each class more and making these classes' representations more dissimilar.

## 3 Mitigating the Dilemma of Few-Shot Class-Incremental Learning

In this section, we first analyze the cause of the dilemma in margin-based classification based on the above interpretation, then we propose our method (named as Class-Level Overfitting Mitigation, CLOM) to mitigate the CO dilemma based on the analysis, as shown in Fig. 5.

### 3.1 Analysis on the Cause of Class-Level Overfitting

Based on the above analysis, we can find the learning of negative-margin-based patterns loosely constrains the given pattern to be shared by both classes. However, such constraint is easily satisfied by ineffective patterns as simple as edges or corners, which could lead to the low discriminability of negative-margin-based patterns. Similarly, the learning of positive-margin-based patterns loosely constrains the given pattern to be specific to the given class. However, such constraint could be satisfied by easily finding patterns sharing no information with other classes, such as finding some complex texture only specific to the given class, which could lead to the low transferability of positive-margin-based patterns. Such **easy-constraint** problems push patterns to be **only** class-specific or shared among classes, making pattern effective in one scenario ineffective in other ones.

To verify the above claims, we analyze how simple or complex the learned patterns are. Inspired by [14], we quantitatively measure the simplicity/complexity of patterns by the similarity between the extracted feature and the simplest feature (e.g., corner or edge features).

We first use the baseline model to train on CIFAR100, and use the first convolutional layer as the simplest feature extractor (denoted as $f_{simple}$), since many works (e.g., [27]) has shown that the first convolutional layer tends to capture corners or edges. Then, we train models with different

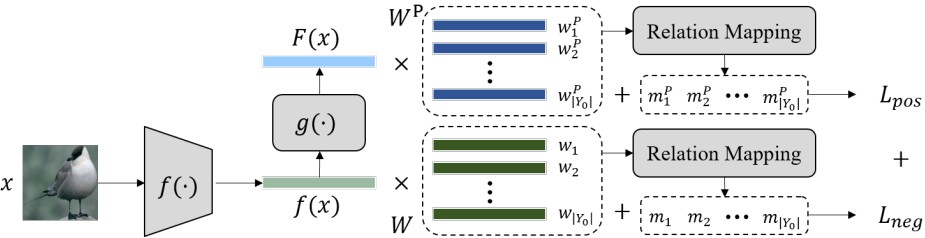

Figure 5: Method (CLOM) framework. We construct the positive-margin-based (PM, $F(\cdot)$) feature from the negative-margin-based (NM, $f(\cdot)$) feature, and map class relations to form a set of class-specific margins, which can effectively mitigate the dilemma of class-level overfitting by providing extra constraint to the NM/PM pattern learning through the learning of PM/NM patterns respectively.

margins, and use the backbone network for feature extraction (denoted as $f_{target}$). After that, we extract $f_{simple}$ and $f_{target}$ features from all images in base classes. Finally, we compare the CKA similarity [14] for measuring the similarity between $f_{simple}$ and the $f_{target}$.

For a sanity check, we first report the similarity between different layers within the baseline model.

Table 1: Sanity check for the CKA measure.

| $f_{target}$ | Conv1-output | Stage1-output | Stage2-output | second-last-Conv | backbone-output |
|---|---|---|---|---|---|
| CKA | 1.0 | 0.8876 | 0.5664 | 0.2097 | 0.1306 |

We can see that the shallower the layer is, the higher the CKA similarity would be, which means the more similar they are to the $f_{simple}$, i.e. the patterns are simpler, more transferable but less discriminative [26]. Then, we report the comparison of the CKA similarity between $f_{simple}$ and $f_{target}$ (backbone feature) of the baseline model trained with margins.

Table 2: CKA between the $f_{simple}$ and baseline backbone features trained with different margins.

| Margin | -0.5 | -0.4 | -0.3 | -0.2 | -0.1 | 0.0 | 0.1 | 0.2 | 0.3 | 0.4 |
|---|---|---|---|---|---|---|---|---|---|---|
| CKA | 0.2432 | 0.2245 | 0.2010 | 0.1661 | 0.1510 | 0.1306 | 0.1149 | 0.0837 | 0.0642 | 0.0576 |

We can see that by applying a negative margin, the CKA similarity clearly increases (even larger than that of the second last convolutional layer when margin < -0.3), showing that the backbone network captures patterns more similar to simplest ones such as edges or corners, which verifies that the model tends to learn simple patterns that are easily shared between classes. When applying a positive margin, the captured patterns grow to be more complex and tend to overfit base classes, making the CKA much smaller than baseline model's backbone-output, which verifies that the model tends to learn complex patterns that are easily specific to a given base class.

Therefore, the key to mitigating the CO dilemma is to extra constrain the pattern learning process.

### 3.2 Mitigating Class-Level Overfitting by Providing Extra Constraint

Negative-Margin-based (NM) patterns are more transferable and class-shared, while Positive-Margin-based (PM) patterns are discriminative and class-specific. These characteristics are similar to the behavior of features from the shallow and deep layers of deep networks [26, 32]. Generally, shallow-layer features encode low-level patterns that are easily shared by most objects therefore more transferable, such as edge or corner, while the deep-layer features encode high-level patterns that are semantically related to only few object classes therefore more discriminative. Such similarity inspires us to view PM patterns as high-level patterns while view NM patterns as (relatively) low-level patterns, and build PM patterns from the NM patterns, just like building high-level features from low-level features in deep networks. As the learning of NM patterns is influenced by PM patterns, this design could benefit the learning of NM patterns by providing extra constraint from the learning of PM patterns, and vice versa, which could therefore handle the easy-constraint problem.

Specifically, given an feature extractor $f(\cdot)$, we add one more layer to construct the PM feature as

$$F(x_i) = g(f(x_i)), \tag{3}$$

where $F(\cdot)$ is the PM feature extractor, $g(\cdot)$ is typically composed of a fully-connected layer, a batch-normalization layer [12] and an activation layer. During training, two classification loss is applied to both $F(\cdot)$ and $f(\cdot)$ with positive and negative margins respectively, represented as

$$L(x_i, y_i) = L_{neg}(x_i, y_i) + L_{pos}(x_i, y_i) \tag{4}$$

$$= -log\frac{e^{\tau(w_{y_i}f(x_i)-m_{neg})}}{e^{\tau(w_{y_i}f(x_i)-m_{neg})} + \sum_{j \neq y_i}e^{\tau w_j f(x_i)}} - \lambda \cdot log\frac{e^{\tau(w_{y_i}^P F(x_i)-m_{pos})}}{e^{\tau(w_{y_i}^P F(x_i)-m_{pos})} + \sum_{j \neq y_i}e^{\tau w_j^P F(x_i)}}, \tag{5}$$

where $w_j^P \in R^{1 \times d^P}$ denotes the classifier weights for $F(\cdot)$ corresponding to class $j$, $m_{neg}$ denotes the negative margin for $f(\cdot)$, $m_{pos}$ is the positive margin for $F(\cdot)$, and $\lambda$ is typically set to 1.0.

Given this modification, the NM patterns $f(\cdot)$ are utilized to construct the discriminative PM patterns $F(\cdot)$. Since $g(\cdot)$ is relatively too simple to capture complex patterns, $f(\cdot)$ will be pushed to be discriminative (validated in Fig. 7), instead of casually learning ineffective patterns shared between similar classes. Similarly, as $F(\cdot)$ are built from the transferable $f(\cdot)$ by the simple $g(\cdot)$, the transferability could be maintained by pushing PM patterns to fit the corresponding base class by transferable information, which improves the transferability of $F(\cdot)$ (validated in Fig. 8).

During testing, given the improved $f(\cdot)$ and $F(\cdot)$, the final feature would be their concatenation.

### 3.3 Further Mitigation of Class-Level Overfitting by Integrating Class Relations

Based on the architecture design, we propose to further mitigate the CO problem through boosting the discriminability and tranferability of NM and PM patterns by integrating the class relations.

For easy understanding, we first introduce the modification to the classification for $f(\cdot)$, and later introduce that for $F(\cdot)$. To modify the margin-based classification from a single margin to margins related to class relations, we first move the margin from the ground-truth logit to all other logits as

$$L(x_i, y_i) = -log\frac{e^{\tau w_{y_i}f(x_i)}}{e^{\tau w_{y_i}f(x_i)} + \sum_{j \neq y_i}e^{\tau(w_j f(x_i)+m(A_{ij}))}}, \tag{6}$$

where $A$ is the adjacency matrix between all classes, $m(\cdot)$ maps the adjacency value between two classes to a margin. Given this modification, if the margin is set to the original fixed value as Eq. 2, the decision boundary between class $y_i$ and other classes remains the same, therefore the new loss in Eq. 6 could be an approximation of the original loss in Eq. 2.

For measuring class relations, we choose to utilize the adjacency matrix of all classes. As both the feature and the classifier's weights are $L_2$ normalized, the adjacency can be measured by the cosine similarity between every two class prototypes as $A_{ij} = cos(P_i, P_j)$, where $P_i$ is the prototype for class $i$, which is typically set to $w_i$ of the classifier.

If two classes are identical, the cosine similarity would reach its upper bound to 1.0. Meanwhile, the average of class relations reflects a global margin that is effective for most classes. Therefore, we design to interpolate the margin from a global effective value to a pre-defined upper bound value as

$$m(A_{ij}) = m_{ave} + \frac{m_{upper} - m_{ave}}{1.0 - A_{ave}} \cdot (A_{ij} - A_{ave}), \tag{7}$$

where $m_{upper}$ is a hyper-parameter controlling the margin for the upper-bound class relation (i.e., two identical classes), $m_{ave}$ is another hyper-parameter controlling the margin for the average class relations, set to the same value as that in Eq. 2, and $A_{ave}$ is the average class relations calculated as $A_{ave} = \frac{1}{|Y_0| \cdot (|Y_0|-1)} \sum_{j=1}^{j=|Y_0|} \sum_{k \neq j} A_{jk}$.

For $F(\cdot)$, we adopt the same modification as $f(\cdot)$, which replaces $f(\cdot)$ with $F(\cdot)$ in Eq. 6, and replaces $A$ with $A^P$ in Eq. 6 where $A_{ij}^P = cos(w_i^P, w_j^P)$. Therefore, the final training objective is

$$L(x_i, y_i) = -log\frac{e^{\tau w_{y_i}f(x_i)}}{e^{\tau w_{y_i}f(x_i)} + \sum\limits_{j \neq y_i}e^{\tau(w_j f(x_i)+m_n(A_{ij}))}} - \lambda \cdot log\frac{e^{\tau w_{y_i}^P F(x_i)}}{e^{\tau w_{y_i}^P F(x_i)} + \sum\limits_{j \neq y_i}e^{\tau(w_j^P F(x_i)+m_p(A_{ij}^P))}}, \tag{8}$$

where $m_p(\cdot)$ and $m_n(\cdot)$ are two interpolate functions as Eq. 7 with positive and negative hyper-parameters $(m_{upper}^P, m_{ave}^P)$ and $(m_{upper}, m_{ave})$ respectively.

In experiments (Fig. 9), we find it beneficial to have $m_{upper} < m_{ave}$ and $m_{upper}^P > m_{ave}^P$, which means for classes with higher similarities, the relation mapping module enables the model to learn more shared (transferable) patterns by applying smaller negative margins, and learn more class-specific (discriminative) patterns by applying larger positive margins. Moreover, due to the connection between NM and PM patterns, the improved transferability in NM patterns would be transmitted to PM patterns and vice versa, as validated in Fig. 9, which further mitigates the CO dilemma.

Table 3: Evaluation datasets.

| Dataset | Total Classes | Base Classes | Novel Classes | Incremental Sessions | Novel-Class Shot | Input Size |
|---|---|---|---|---|---|---|
| CIFAR100 | 100 | 60 | 40 | 8 | 5 | $32 \times 32$ |
| CUB200 | 200 | 100 | 100 | 10 | 5 | $224 \times 224$ |
| *mini*ImageNet | 100 | 60 | 40 | 8 | 5 | $84 \times 84$ |

Table 4: Comparison with state-of-the-art works on the CUB200 dataset.

| Method | S0 | S1 | S2 | S3 | S4 | S5 | S6 | S7 | S8 | S9 | S10 |
|---|---|---|---|---|---|---|---|---|---|---|---|
| Finetune | 68.68 | 43.70 | 25.05 | 17.72 | 18.08 | 16.95 | 15.10 | 10.06 | 8.93 | 8.93 | 8.47 |
| Rebalancing [11] | 68.68 | 57.12 | 44.21 | 28.78 | 26.71 | 25.66 | 24.62 | 21.52 | 20.12 | 20.06 | 19.87 |
| iCaRL [18] | 68.68 | 52.65 | 48.61 | 44.16 | 36.62 | 29.52 | 27.83 | 26.26 | 24.01 | 23.89 | 21.16 |
| EEIL [2] | 68.68 | 53.63 | 47.91 | 44.20 | 36.30 | 27.46 | 25.93 | 24.70 | 23.95 | 24.13 | 22.11 |
| TOPIC [22] | 68.68 | 62.49 | 54.81 | 49.99 | 45.25 | 41.40 | 38.35 | 35.36 | 32.22 | 28.31 | 26.26 |
| Decoupled-NegCosine [16] | 74.96 | 70.57 | 66.62 | 61.32 | 60.09 | 56.06 | 55.03 | 52.78 | 51.50 | 50.08 | 48.47 |
| CEC [28] | 75.85 | 71.94 | 68.50 | 63.50 | 62.43 | 58.27 | 57.73 | 55.81 | 54.83 | 53.52 | 52.28 |
| FSLL+SS [17] | 75.63 | 71.81 | 68.16 | 64.32 | 62.61 | 60.10 | 58.82 | 58.70 | 56.45 | 56.41 | 55.82 |
| FACT [30] | 75.90 | 73.23 | 70.84 | 66.13 | 65.56 | 62.15 | 61.74 | 59.83 | 58.41 | 57.89 | 56.94 |
| IDLVQ-C [4] | 77.37 | 74.72 | 70.28 | 67.13 | 65.34 | 63.52 | 62.10 | 61.54 | 59.04 | 58.68 | 57.81 |
| CLOM (Ours) | **79.57** | **76.07** | **72.94** | **69.82** | **67.80** | **65.56** | **63.94** | **62.59** | **60.62** | **60.34** | **59.58** |

After the base-session training, the feature extractor $F(\cdot)$ and $f(\cdot)$ will be applied to the training data of each session to obtain the extended classifier weight $W = \{w_1^0, w_2^0, ..., w_{|Y_0|}^0\} \cup ... \cup \{w_1^k, ..., w_{|Y_k|}^k\}$ by averaging extracted features, and the final performance of each session will be obtained based on the classification of all the encountered classes' test samples.

## 4 Experiments

In this section, we first introduce the experiment settings, then compare the proposed method with the state-of-the-art methods, and finally report the ablation study for the effectiveness of each design.

### 4.1 Datasets

Datasets include CIFAR100 [15], Caltech-UCSD Birds-200-2011 (CUB200) [24] and *mini*ImageNet [23] as listed in Tab. 3 following the split in [22]. For details, please refer to the appendix.

### 4.2 Implementation Details

The implementation is based on CEC's code [28], and our code will be released[3]. For CIFAR100, we set $d^P$=256, set $m_{ave}$=-0.2, set $m_{upper}$=-0.5, and we have $m_{ave}^P$=0.1 and $m_{upper}^P$=0.2. For CUB200, we scale the learning rate of the backbone network to 10% of the global learning rate since the pre-training of the backbone is adopted [30, 28], and set $d^P$ to 8192. Then we have $m_{ave}$=-0.2 and $m_{upper}$=-0.25 and $m_{ave}^P$=0.3 and $m_{upper}^P$=0.6. For *mini*ImageNet, we set $d^P$ to 4096, and have $m_{ave}$=-0.2 and $m_{upper}$=-0.5 and $m_{ave}^P$=0.1 and $m_{upper}^P$=0.2. Please refer to appendix for details.

### 4.3 Comparison with the State-of-the-Art

Comparisons with the state-of-the-art works are listed in Tab. 4 and Fig. 6, where we can achieve state-of-the-art performance on all three datasets. Specifically, we can first see that our method, as a prototype-based method (e.g., CEC [28]), could significantly outperform finetune-based methods (e.g., iCaRL [18]). This is because the few-shot training data could not provide sufficient information for novel-class learning, therefore directly freezing parameters on novel classes could reduce the catastrophic-forgetting problem brought by the finetuning. Our method also outperforms other prototype-based methods, this is because the core of the prototype-based methods lies in the metric learning [20]. As empirically proved by current works [21, 25, 8, 16], applying the margin-based classification could effectively improve the embedding space learned by metric-based methods. Since our method handles the difficulty of applying margin-based classification (i.e., class-level overfitting) to the FSCIL task, our method could beat these prototype-based ones. For detailed numbers of CIFAR100 and *mini*ImageNet, please refer to the appendix.

### 4.4 Ablation Study of Modules

The ablation study of each module is reported in Tab. 5, where $g(\cdot)$ denotes adding another layer, *Margin* denotes the applicant of margin-based classification, and *Relation* refers to the relation map-

---

[3]https://github.com/Zoilsen/CLOM

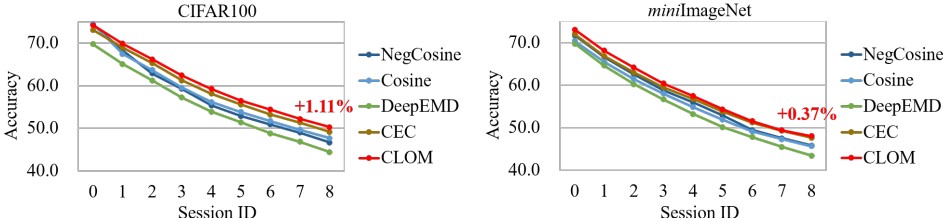

Figure 6: Comparison with state-of-the-art works on CIFAR100 and *mini*ImageNet.

Table 5: Ablation study of modules on the last incremental session of three datasets.

| Method | CUB200 | | | CIFAR100 | | | *mini*ImageNet | | |
|---|---|---|---|---|---|---|---|---|---|
| | Overall | Novel | Base | Overall | Novel | Base | Overall | Novel | Base |
| Baseline | 57.78 | 45.97 | 79.48 | 47.02 | 37.40 | 72.32 | 46.58 | 31.02 | 72.33 |
| + $g(\cdot)$ | 57.21 | 47.13 | 79.39 | 48.37 | 39.55 | 72.70 | 46.79 | 30.97 | 72.60 |
| + Margin | 58.73 | 49.93 | 79.47 | 49.21 | 40.22 | 73.72 | 47.30 | 32.07 | 72.93 |
| + Relation | **59.58** | **50.89** | **79.57** | **50.25** | **41.17** | **74.20** | **48.00** | **33.60** | **73.08** |

ping of the margin. We study from three aspects: *Base*-class, *Novel*-class, and *Overall* accuracy of the last incremental session. From Tab. 5, we can see that

• *Simply applying another layer cannot consistently improve the performance*.

• *The designed architecture could mitigate class-level overfitting*. Compared with experiments in Fig. 2 where no *Overall* performance improvements can be obtained by simply adding margins, the performance here is clearly improved by adding margins on the designed architecture. Moreover, the overall improvements originate from not only the improved *Base* performance, but also the boosted *Novel* performance, demonstrating the mitigation of class-level overfitting (CO) problem.

• *Relation mapping could further improve performance by mitigating class-level overfitting*.

### 4.5 Verification of Class-Level Overfitting Mitigation

**Mitigating the class-level overfitting by the architecture design.** We first compare our method with the baseline method which applies the margin directly to the backbone feature in Fig. 7 and 8, so as to validate the mitigation of CO. In Fig. 7, we apply the negative margin to the backbone (NM) feature while applying no margin to the PM feature, and see the accuracy of the NM feature against the baseline. As can be seen, while keeping the improvements on novel classes (mid), our method (orange) maintains the base-class accuracy (right), in contrast to a sharp decrease of the baseline (blue) when the margin decreases, which makes our NM feature significantly outperforms the baseline in terms of the overall accuracy (left). Similarly, in Fig. 8, our PM feature also maintains a higher performance on novel classes when the margin increases. These experiments validate the mitigation of the CO dilemma: by implicitly guiding the learning of the PM and NM features by each other, the transferability or discriminability drop on them can be mitigated respectively.

**Further mitigating class-level overfitting by relation mapping.** To validate the relation mapping module, in Fig. 9 we fixed the average margin (i.e., $m_{ave}$=-0.2 in Fig. 9.1st and $m_{ave}^P$=0.3 in Fig. 9.2nd) and experiment with different upper margins (i.e., $m_{upper}$ in Fig. 9.1st and $m_{upper}^P$ in Fig. 9.2nd). Similar to Fig. 7, when conducting experiments on one branch, the hyper-parameters of the other branch are fixed. We can see the relation mapping module can indeed help the learning, as the upper margins are significantly different from the average margin. We also plot the NM and PM performance on base classes with negative margins in Fig. 9(3rd), and those on novel classes with positive margins in Fig. 9(4th). We can see the NM and PM performance are improved simultaneously, which validates the improvements brought by the extra guidance of each features. Therefore, the relation mapping module can also help to mitigate the class-level overfitting problem by building PM features with more transferable patterns, and vice versa. Moreover, it is interesting to find the performance is improved when $m_{upper} < m_{ave}$ and $m_{upper}^P > m_{ave}^P$, which means for classes with higher similarities, the relation mapping module enables the model to learn more shared patterns by applying negative margins with larger absolute value, and learn more class-specific patterns by applying larger positive margins. Also, this is consistent with the results in Fig. 3.

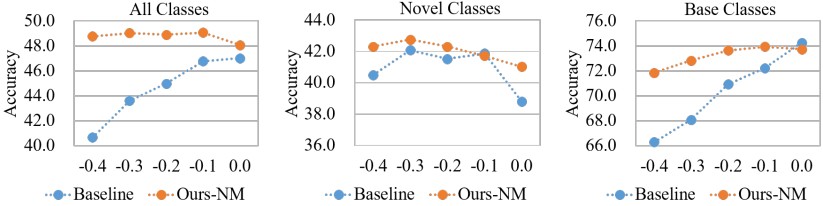

Figure 7: NM feature's accuracy compared with the baseline with negative margins (CIFAR100).

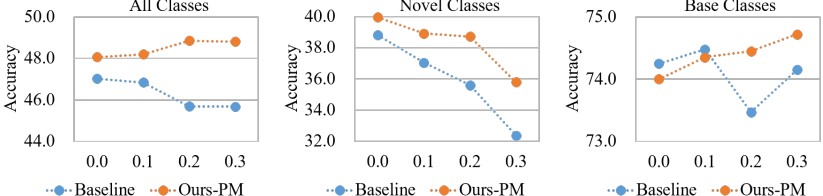

Figure 8: PM feature's accuracy compared with the baseline with positive margins (CIFAR100).

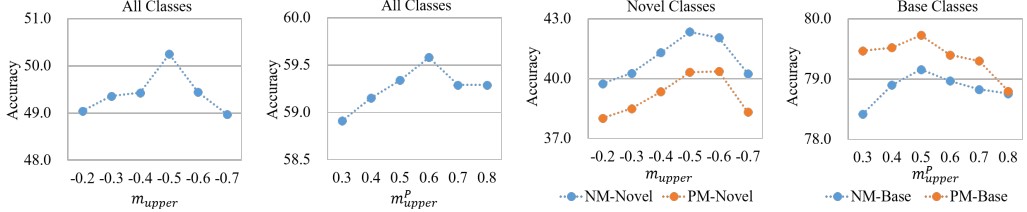

Figure 9: Upper margins in the relation mapping module ($m_{upper}$: CIFAR100, $m_{upper}^P$ : CUB200).

### 4.6 Mitigating the Easy-Constraint Problem

Following experiments in Tab. 1, we also utilize the CKA similarity between the simplest feature $f_{simple}$ and the NM/PM feature to validate the mitigation of the easy-constraint problem. By applying our method, for the NM (or PM) patterns, we set $f_{target}$ to $f(\cdot)$ (or $F(\cdot)$) in Fig.5 and set the positive/negative margin to 0.3/-0.4, and report the CKA below.

Table 6: CKA between $f_{simple}$ and NM (left) or PM (right) features trained with different margins.

| Margin | -0.5 | -0.4 | -0.3 | -0.2 | -0.1 | 0.0 |
|---|---|---|---|---|---|---|
| CKA | 0.1867 | 0.1779 | 0.1724 | 0.1638 | 0.1552 | 0.1427 |

| Margin | 0.0 | 0.1 | 0.2 | 0.3 | 0.4 |
|---|---|---|---|---|---|
| CKA | 0.1476 | 0.1439 | 0.1430 | 0.1214 | 0.1100 |

In Tab. 6 (left), compared with Tab. 2, CKA decreases clearly compared with the baseline method, which validates that NM patterns learned by our method are more complex and less similar to edges and corners than the baseline method, verifying the mitigation of the easily-satisfied constraint problem by extra supervision from the learning of PM patterns. Also, in Tab. 6 (right), compared with Tab. 2, CKA is much larger than that of the baseline method, which validates that the PM patterns are more similar to the simplest patterns than the baseline method, which makes it less overfitting the base classes, verifying the mitigation by extra supervision from the learning of NM patterns.

## 5 Conclusion

In this paper, we focus on the dilemma in the margin-based classification for FSCIL. We first interpret the dilemma as a class-level overfitting problem from the aspect of pattern learning, then find the cause of this problem lies in the easily-satisfied constraint of learning shared or class-specific patterns. Based on the analysis, we design a method (CLOM) to mitigate the dilemma by constructing PM patterns from NM patterns, and mapping class relations into class-specific patterns. Extensive experiments on three public datasets validate the effectiveness and outstanding performance of the proposed method.

## Acknowledgements

This work is supported by National Natural Science Foundation of China under grants U1836204, U1936108, 62206102, Science and Technology Support Program of Hubei Province under grant 2022BAA046, and 2022 CCF-DiDi Gaiya Young Scholar Research Fund.

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
