# Margin-Based Few-Shot Class-Incremental Learning with Class-Level Overfitting Mitigation

**Yixiong Zou**[1], **Shanghang Zhang**[2], **Yuhua Li**[1] **and Ruixuan Li**[1*]

[1]School of Computer Science and Technology, Huazhong University of Science and Technology
[2]School of Computer Science, Peking University

[1]{yixiongz, idcliyuhua, rxli}@hust.edu.cn, [2]shanghang@pku.edu.cn

## A    Appendix for Related Work

**Few-Shot Class-Incremental Learning.**    Few-shot class-incremental learning (FSCIL) can be roughly divided into finetune-based methods [16, 26, 4, 31] and metric-based methods [40, 44]. The former finetunes the model on the novel-class training data, focusing on avoiding catastrophic forgetting during the finetuning. The latter achieves this goal by freezing parameters of the pre-trained model, and recognize the novel classes by the prototype-based [28] Nearest-Neighbor classification, which shares the same concept with the metric-based few-shot learning works [32, 28, 30]. This paper can be categorized into the metric-based FSCIL methods.

**Few-Shot Learning.**    Few-shot Learning (FSL) [32, 28] focuses on the recognition of novel classes with only few training samples. It can be roughly divided into metric-based methods [32, 28, 39, 25], meta-learning based methods [12, 23, 27] and augmentation-based methods [36, 14, 1]. Metric-based methods share the concept with those of FSCIL, which aims to learn a good embedding space to recognize novel classes. Therefore, methods [25, 21, 39] effective in FSL can also be effective in FSCIL. However, as the original FSL task (except for some subspecies task of FSL, e.g., generalized FSL [13]) does not require the recognition of base classes, FSL generally emphasizes more on the novel-class generalization.

**Margin-Based Classification.**    Margin-based classification [29, 34, 11, 21] has been widely utilized in metric learning. For example, [34] and [11] proposed to add a positive margin to the classification to learn a better fine-grained embedding space for face recognition. [21] proposed to add a negative margin to benefit the novel-class recognition. However, the reason why the model behaves differently on base classes and novel classes has not been fully studied, and this paper tackles this problem from the aspect of pattern learning.

**Discriminability vs. Transferability.**    The dilemma between transferability and discriminability has been researched in various research domains [7, 8, 9]. For example, [7] studied this problem for adversarial domain adaptation, [8] studied this problem from the aspect of information-bottleneck theory, and [9] investigated this problem under the label insufficient situations. However, it still remains to be studied under the metric-based FSCIL or FSL scenario, and this paper study this problem from the aspect of pattern learning.

**Semantic Pattern Learning.**    Interpretability of deep networks [43, 2] shows that each channel in the extracted feature can be understood as a pattern extractor, which can be used to dissect [2] the given network for the encoded knowledge. Moreover, each convolution kernel, as a pattern extractor, can be understood as a semantic template [5], and the activation on each channel can be viewed as the matching score between the template and the input. Based on these studies, [45] proposed to

---

[*]Corresponding author.

36th Conference on Neural Information Processing Systems (NeurIPS 2022).

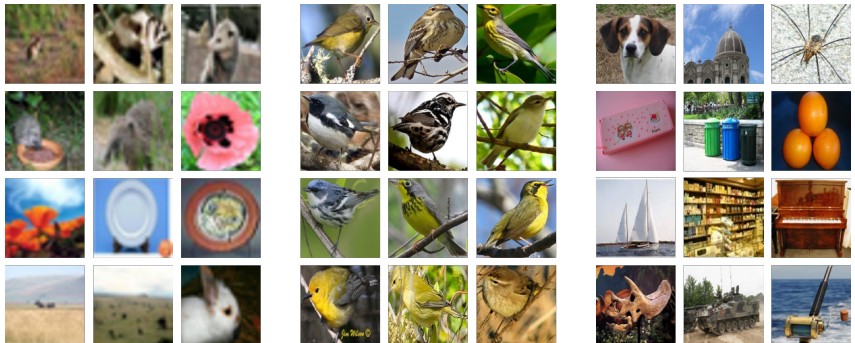

Figure 1: Samples of CIFAR100 (left), CUB200 (mid) and *mini*ImageNet (right).

Table 1: Comparison with state-of-the-art works on the CUB200 dataset.

| Method | S0 | S1 | S2 | S3 | S4 | S5 | S6 | S7 | S8 | S9 | S10 |
|---|---|---|---|---|---|---|---|---|---|---|---|
| Finetune | 68.68 | 43.70 | 25.05 | 17.72 | 18.08 | 16.95 | 15.10 | 10.06 | 8.93 | 8.93 | 8.47 |
| Rebalancing [16] | 68.68 | 57.12 | 44.21 | 28.78 | 26.71 | 25.66 | 24.62 | 21.52 | 20.12 | 20.06 | 19.87 |
| iCaRL [26] | 68.68 | 52.65 | 48.61 | 44.16 | 36.62 | 29.52 | 27.83 | 26.26 | 24.01 | 23.89 | 21.16 |
| EEIL [4] | 68.68 | 53.63 | 47.91 | 44.20 | 36.30 | 27.46 | 25.93 | 24.70 | 23.95 | 24.13 | 22.11 |
| TOPIC [31] | 68.68 | 62.49 | 54.81 | 49.99 | 45.25 | 41.40 | 38.35 | 35.36 | 32.22 | 28.31 | 26.26 |
| Decoupled-NegCosine [21] | 74.96 | 70.57 | 66.62 | 61.32 | 60.09 | 56.06 | 55.03 | 52.78 | 51.50 | 50.08 | 48.47 |
| CEC [40] | 75.85 | 71.94 | 68.50 | 63.50 | 62.43 | 58.27 | 57.73 | 55.81 | 54.83 | 53.52 | 52.28 |
| FSLL+SS [22] | 75.63 | 71.81 | 68.16 | 64.32 | 62.61 | 60.10 | 58.82 | 58.70 | 56.45 | 56.41 | 55.82 |
| FACT [44] | 75.90 | 73.23 | 70.84 | 66.13 | 65.56 | 62.15 | 61.74 | 59.83 | 58.41 | 57.89 | 56.94 |
| IDLVQ-C [6] | 77.37 | 74.72 | 70.28 | 67.13 | 65.34 | 63.52 | 62.10 | 61.54 | 59.04 | 58.68 | 57.81 |
| Ours | **79.57** | **76.07** | **72.94** | **69.82** | **67.80** | **65.56** | **63.94** | **62.59** | **60.62** | **60.34** | **59.58** |

view each class as a composition of semantic patterns. Based on the above previous works, we can analyze the model behavior from the aspect of pattern learning.

# B    Appendix for Experiments

## B.1    Detailed Dataset Description

**CIFAR100.**    CIFAR100 [19] is a challenging dataset consisting of 100 classes and 60,000 images with the shape of $32 \times 32$ as shown in Fig. 1 (left). We adopt this dataset with consent from the authors [19]. For each class, there are 500 images for training and 100 images for testing. As split in [31], 60 classes are chosen as the base-session classes, and 40 classes are used as novel classes. The 40 novel classes are further divided into 8 incremental sessions where each session has 5 classes with 5 training samples in each class for training.

**Caltech-UCSD Birds-200-2011 (CUB200).**    The CUB200 [33] dataset is designed for the fine-grained classification of birds as shown in Fig. 1 (mid). We adopt this dataset with consent from the authors [33]. It contains 11,788 images from 200 classes. As split in [31], 100 classes are chosen as the base-session classes and the remaining are the novel classes. The 100 novel classes are further divided into 10 incremental sessions where each session contains 10 classes with 5 training samples in each class.

***mini*ImageNet.**    The *mini*ImageNet [32] dataset is a subset of the ImageNet [10] dataset, containing 100 classes and 600 images in each class, and the images of it are resized to $84 \times 84$ as shown in Fig. 1 (right). We adopt this dataset with consent from the authors [32]. We follow [31] to split it into 60 base classes and 40 novel classes, and construct 8 incremental sessions from the 40 novel classes, where each session contains 5 classes with 5 training samples in each class.

## B.2    Detailed Implementation Details

Our implementation is based on the code released by CEC [40] under the MIT license.

Table 2: Comparison of state-of-the-art works on the CIFAR100 dataset.

| Method | S0 | S1 | S2 | S3 | S4 | S5 | S6 | S7 | S8 |
|---|---|---|---|---|---|---|---|---|---|
| Finetune | 64.10 | 39.61 | 15.37 | 9.80 | 6.67 | 3.80 | 3.70 | 3.14 | 2.65 |
| Pre-Allocated RPC [24] | 64.50 | 54.93 | 45.54 | 30.45 | 17.35 | 14.31 | 10.58 | 8.17 | 5.14 |
| iCaRL [26] | 64.10 | 53.28 | 41.69 | 34.13 | 27.93 | 25.06 | 20.41 | 15.48 | 13.73 |
| EEIL [4] | 64.10 | 53.11 | 43.71 | 35.15 | 28.96 | 24.98 | 21.01 | 17.26 | 15.85 |
| Rebalancing [16] | 64.10 | 53.05 | 43.96 | 36.97 | 31.61 | 26.73 | 21.23 | 16.78 | 13.54 |
| TOPIC [31] | 64.10 | 55.88 | 47.07 | 45.16 | 40.11 | 36.38 | 33.96 | 31.55 | 29.37 |
| Decoupled-NegCosine [21] | 74.36 | 68.23 | 62.84 | 59.24 | 55.32 | 52.88 | 50.86 | 48.98 | 46.66 |
| Decoupled-Cosine [32] | **74.55** | 67.43 | 63.63 | 59.55 | 56.11 | 53.80 | 51.68 | 49.67 | 47.68 |
| Decoupled-DeepEMD [39] | 69.75 | 65.06 | 61.20 | 57.21 | 53.88 | 51.40 | 48.80 | 46.84 | 44.41 |
| CEC [40] | 73.07 | 68.88 | 65.26 | 61.19 | 58.09 | 55.57 | 53.22 | 51.34 | 49.14 |
| CLOM (Ours) | 74.20 | **69.83** | **66.17** | **62.39** | **59.26** | **56.48** | **54.36** | **52.16** | **50.25** |

Table 3: Comparison of state-of-the-art works on the *mini*ImageNet dataset.

| Method | S0 | S1 | S2 | S3 | S4 | S5 | S6 | S7 | S8 |
|---|---|---|---|---|---|---|---|---|---|
| Finetune | 61.31 | 27.22 | 16.37 | 6.08 | 2.54 | 1.56 | 1.93 | 2.60 | 1.40 |
| Pre-Allocated RPC [24] | 61.25 | 31.93 | 18.92 | 13.90 | 14.37 | 15.57 | 16.15 | 12.33 | 12.28 |
| iCaRL [26] | 61.31 | 46.32 | 42.94 | 37.63 | 30.49 | 24.00 | 20.89 | 18.80 | 17.21 |
| EEIL [4] | 61.31 | 46.58 | 44.00 | 37.29 | 33.14 | 27.12 | 24.10 | 21.57 | 19.58 |
| Rebalancing [16] | 61.31 | 47.80 | 39.31 | 31.91 | 25.68 | 21.35 | 18.67 | 17.24 | 14.17 |
| TOPIC [31] | 61.31 | 50.09 | 45.17 | 41.16 | 37.48 | 35.52 | 32.19 | 29.46 | 24.42 |
| Decoupled-NegCosine [21] | 71.68 | 66.64 | 62.57 | 58.82 | 55.91 | 52.88 | 49.41 | 47.50 | 45.81 |
| Decoupled-Cosine [32] | 70.37 | 65.45 | 61.41 | 58.00 | 54.81 | 51.89 | 49.10 | 47.27 | 45.63 |
| Decoupled-DeepEMD [39] | 69.77 | 64.59 | 60.21 | 56.63 | 53.16 | 50.13 | 47.79 | 45.42 | 43.41 |
| CEC [40] | 72.00 | 66.83 | 62.97 | 59.43 | 56.70 | 53.73 | 51.19 | 49.24 | 47.63 |
| CLOM (Ours) | **73.08** | **68.09** | **64.16** | **60.41** | **57.41** | **54.29** | **51.54** | **49.37** | **48.00** |

**CIFAR100.** We follow CEC [40] to utilize ResNet20 [15] as the backbone network. The data augmentation includes regular augmentation techniques, i.e., the random resized crop, the random horizontal flip and the normalization of images. Note that for fair comparison, we do not utilize the auto-augment tricks as done in [44]. We follow CEC to train the model for 100 epochs, and decay the original learning rate 0.1 to 0.01 and 0.001 at the 60th and 70th epoch respectively. Other details have been illustrated in the paper.

**CUB200.** ResNet18 [15] is utilized as the backbone network following CEC. Also, the pre-training from ImageNet is adopted as CEC and TOPIC [31]. Therefore, we shrink the learning rate of the backbone network to 0.01, while keeping the that for the global learning rate to be 0.1. The model is trained for 80 epochs, and the learning rate decay is conducted at the 40th and 50th epoch. Data augmentation techniques are the same as those in CIFAR100. Other details have been illustrated in the paper.

***mini*ImageNet.** ResNet18 is also utilized as the backbone network following CEC. Unlike that in CUB200, the backbone is trained from scratch. The data augmentation is also the same as that on CIFAR100, where the auto-augment is not adopted neither. The model is trained for 180 epochs, and the learning rate is decayed to 10% at the 90th and the 120th epoch. Other details have been illustrated in the paper.

## B.3 Detailed Incremental Performance

In the paper, we plot the comparison on CUB200 and *mini*ImageNet by the performance curve. For clarity, we also report the exact numbers in Tab. 2 and 3, and attach the results on CUB200 in Tab. 1 for easy comparison.

From these tables, we can see that our performance measured by the last incremental session is the highest, even with lower base-class performance as in Tab. 2. This result verifies that our model could achieve better novel-class generalization without harming the base-class performance, i.e., mitigating the class-level overfitting problem.

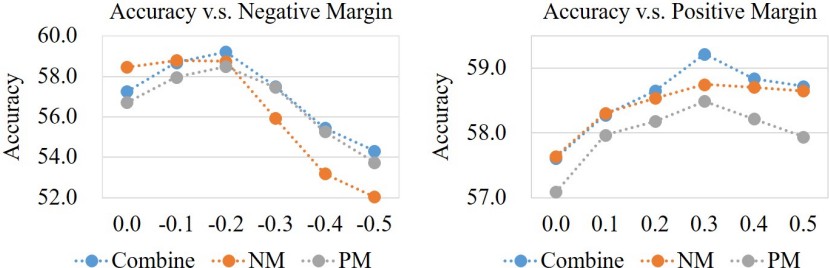

Figure 2: Performance of the NM feature, the PM feature and the combined feature w.r.t. the negative (left) and positive (right).

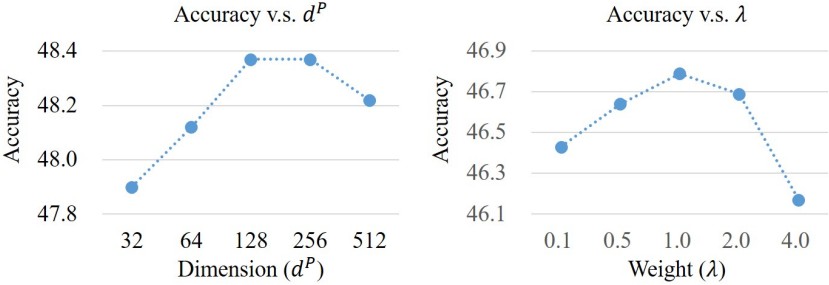

Figure 3: Left: sensitivity study of the PM feature dimension ($d^P$). Right: sensitivity study of the weight for PM feature classification ($\lambda$).

## B.4 Extra Ablation Study

**Pattern Cooperation** To verify the learning of positive- and negative-margin-based features, we study the performance on different features w.r.t. margins without the relation mapping module. Results are reported in Fig. 2 (left) and Fig. 2 (right), where *NM* denotes the negative-margin-based feature, *PM* positive-margin-based feature, and *Combine* refers to the concatenation of two features. The experiments are conducted with the margin of the other branch fixed. We can see that

**(1) Branches learn to encode different information as the difference of margin between two features increases**. As can be seen, when the margin is 0.0, the performance of the combined feature is lower than one of the branches. As the margin decreases (increases) in the left (right) figure, the performance of the combined feature begin to surpass both two branches, which validates that two branches learn to encode different information (i.e., one for transferability and one for discriminability) as the difference of margins increases.

**(2) The improvements on one branch can also help the learning of the other branch**, as the performance of two branches increases and decreases synchronously. This is because the PM feature is built from the NM feature, therefore the learning of one branch can implicitly help the other one.

**Sensitivity Study** To verify the choice of hyper-parameters, we plot the sensitivity study of the PM feature's dimension $d^P$, and the weight $\lambda$ of the classification loss for the PM feature. Results are plotted in Fig. 3. We can see that $d^P$ reaches its maximum performance on CIFAR100 at the chosen optimal value, i.e., 256, and drops when keeping increasing it. Other datasets follow the same trend for the chosen values, i.e., 8192 for CUB200 and 4096 for *mini*ImageNet.

Similarly, the overall accuracy reaches the top value when $\lambda$ equals 1.0, which means the contribution from NM and PM features are close. Other datasets also follow the same trend for the chosen values. Moreover, note that the best $\lambda$ for CUB200 is 0.01, this is because the backbone model adopts the pre-training [31, 40] from ImageNet, therefore the guidance from PM patterns on NM patterns should be weakened.

Table 4: Full combinations of different margins without the relation mapping module (CIFAR100).

| Margin | -0.5 | -0.4 | -0.3 | -0.2 | -0.1 | 0.0 | 0.1 | 0.2 | 0.3 | 0.4 |
|---|---|---|---|---|---|---|---|---|---|---|
| -0.5 | 41.16 | 42.51 | 43.30 | 45.68 | 47.93 | 49.33 | 49.25 | 49.27 | 48.53 | 48.01 |
| -0.4 | 42.24 | 42.37 | 44.37 | 45.96 | 48.17 | 48.93 | 48.94 | 48.76 | 49.01 | 47.15 |
| -0.3 | 43.99 | 44.45 | 45.06 | 45.85 | 47.51 | 48.90 | 49.38 | 49.09 | 48.87 | 48.08 |
| -0.2 | 45.05 | 44.71 | 45.61 | 45.85 | 48.20 | 48.08 | 49.60 | 48.65 | 48.53 | 48.08 |
| -0.1 | 47.50 | 47.11 | 46.71 | 46.69 | 47.99 | 48.87 | 48.92 | 49.21 | 48.55 | 47.99 |
| 0.0 | 47.38 | 47.7 | 47.00 | 47.94 | 47.43 | 48.48 | 48.67 | 49.14 | 47.96 | 47.87 |
| 0.1 | 47.32 | 48.14 | 48.01 | 47.96 | 47.95 | 48.37 | 47.62 | 48.02 | 48.35 | 47.32 |
| 0.2 | 47.44 | 46.66 | 47.7 | 47.05 | 47.27 | 47.21 | 47.40 | 47.55 | 47.64 | 46.65 |
| 0.3 | 46.61 | 46.23 | 46.29 | 46.21 | 47.01 | 47.11 | 46.73 | 46.37 | 46.55 | 45.68 |
| 0.4 | 45.37 | 45.35 | 45.81 | 45.70 | 46.26 | 46.69 | 45.79 | 46.51 | 46.08 | 45.56 |

# C  Full combinations of different margins

We report the experiments with all possible combinations of margins on CIFAR100 in Tab. 4, where the horizontal axis represents the margin attached to higher layer $F(\cdot)$, and the vertical axis denotes the margin for the lower layer $f(\cdot)$.

We can see that the margins adopted in the paper (i.e., positive margin on the higher layer + negative margin on the lower layer, the top right area) show clear improvements over the baseline (i.e., margins on both layers are 0.0). Instead, other combinations of margins (e.g., negative margin on the higher layer + positive margin on the lower layer, the left bottom area) show a much lower performance compared with the baseline, which verifies our choice of hyper-parameters and our insight: build positive-margin-based patterns from negative-margin-based patterns.

# D  Comparison with face recognition methods

We implemented some methods in the face recognition community that might be relevant to our work, including

(1) Adaptive-margin-based methods (CurricularFace [17], AdaCos [41], ElasticFace [3], AMR-Loss [42]), such as adjusting the margin value according to the intra/inter-class angles. However, these methods always rely more on the angles across the training time than on angles across all training classes, which differs from our class-relation-based mapping mechanism which rely more on angles across all training classes.

(2) Relational-margin-based methods (TRAML [20]), which maps the class relationship to the margin. However, this work takes the semantic embedding (such as attributes) of each class as input, and utilizes a network to learn the relational margin, which differs from our work in that we do not need further training a network nor the attributes to obtain the relational margin. Moreover, we specifically design the search space of hyper-parameters of the linear mapping to allow similar classes to have a margin with larger absolute value, so that the mapping is more interpretable than mapping by a learnable black-box neural network.

Below we empirically compare our method with the above face recognition methods on CIFAR100 following settings in our paper. Our aim of experiments includes (1) verifying whether they can solve the dilemma between base-class performance and novel-class generalization and (2) verifying whether they can effectively capture the relationship between classes.

From this table, we can see that (1) these methods cannot solve the dilemma by adding directly to the baseline method, since none of them can achieve higher base and novel accuracy simultaneously compared with the baseline performance, while ours (baseline + NM/PM) can; (2) with the NM/PM architecture design, these methods can hardly capture the relationship between classes, since they could not achieve performance significantly higher than the baseline + NM/PM ones, which verify the effectiveness of our method under the few-shot class-incremental learning task.

Table 5: Comparison with face recognition methods on CIFAR100.

| CIFAR100 | overall | novel | base |
|---|---|---|---|
| baseline | 47.02 | 37.40 | 72.32 |
| baseline + CurricularFace | 47.22 | 35.77 | 72.67 |
| baseline + AdaCos | 44.48 | 26.07 | 72.55 |
| baseline + TRAML | 47.31 | 36.32 | 73.15 |
| baseline + ElasticFace | 47.09 | 36.60 | 72.40 |
| baseline + AMR-Loss | 46.66 | 33.37 | 72.58 |
| baseline + NM/PM (ours) | **49.21** | **40.22** | **73.72** |
| baseline + NM/PM + CurricularFace | 49.48 | 40.07 | 74.10 |
| baseline + NM/PM + AdaCos | 45.24 | 27.20 | 73.83 |
| baseline + NM/PM + TRAML | 48.18 | 39.35 | 73.73 |
| baseline + NM/PM + ElasticFace | 49.43 | 40.70 | 74.09 |
| baseline + NM/PM + AMR-Loss | 49.22 | 38.92 | 74.15 |
| baseline + NM/PM + relation (ours) | **50.25** | **41.17** | **74.20** |

# E  Broader Impact

We propose a margin-based FSCIL method to mitigate the class-level overfitting problem in the margin-based classification. This work can also be adopted in fields other than FSCIL, such as FSL, image retrieval [11] and person re-identification [35], because the class-level overfitting (CO) problem handled by this method is not limited to the FSCIL task. The limitation of the work is to omit the many-shot scenarios where the finetuning on novel classes cannot be ignored. However, as the novel-class embedding extracted by our method could provide an effective initialization for the novel-class classifier, this method still has the potential to be further developed for the realistic many-shot scenarios.