# OpenReview forum: "Margin-Based Few-Shot Class-Incremental Learning with Class-Level Overfitting Mitigation"
_NeurIPS.cc/2022/Conference — NeurIPS 2022 Accept_

### Official Review · Reviewer_9Up9 · 2022-07-09

**Rating:** 5
**Confidence:** 5
**Soundness:** 3 good
**Presentation:** 3 good
**Contribution:** 3 good

**Summary:**

This paper deals with the few shot incremental learning task. The authors identify the inferior trade-off between base- and novel-class as a class-level overfitting problem during the margin-based training, and propose a new margin-based learning method with extra constraint to overcome the problem. The experiments show the advantage of the proposed method on certain commonly used benchmarks.

**Questions:**

It is better if the paper includes the extra experiments on face recognition datasets.



**Limitations:**

No obvious negative societal impact observed.

**Strengths And Weaknesses:**

Originality
Pros: Two branches for pos and neg margin setting, respectively, are new and interesting to me.
Cons: The class relation in the inter-class logit is not new, but is studied by many methods in the face recognition community.

Quality
Pros: The overall quality of this paper is good. The method is well formulated, based on the solid observation and analysis. The experiments are clear to show the effectiveness.
Cons: It lacks the comparison with the similar methods in the face recognition community.

Clarity
Pros: The method and the experiments are well presented.
Cons: Missing definition of horizontal and vertical axes in Fig. 2.

Significance
Pros: The proposed method will be useful for practical application of incremental learning task.
Cons: -

---

> ### Author Response · Authors · 2022-08-02
> **Response to Reviewer 9Up9**
>
> We thank the reviewer for the valuable advice. We address each below.
>
> # 1. Novelty compared with face recognition
>
> We would like to emphasize that although both face recognition and our work utilize the concept of margin-based classification, our work differs significantly from face recognition in the following aspect:
>
> (1) Different data distribution: face recognition dataset could be viewed as one of the finest-grained classification datasets, where the classification relies on the very subtle differences between two images. But our few-shot class-incremental learning (FSCIL) task handles data that are more general, such as ImageNet classes, where larger semantic gaps and nature class relations (such as WordNet hierarchical relations) exist between classes. This intrinsically makes the designing concept to be different for these two tasks. For example, for face recognition datasets, patterns (such as eyes, nose, ears) are easier to be shared between classes so that a large positive margin benefits the model, even for the novel face identities. But for our general-class classification task, we need to consider the case where no shared patterns could be learned between classes due to the semantic gap, therefore a negative margin is needed.
>
> (2) Different task: we consider both the base-class performance and novel-class generalization, which is the core of this work.
>
> Specifically, in terms of our method, our difference with current face recognition methods is in
>
> (1) Different view of analysis: our analysis is from the aspect of pattern learning, and find that the class relationship is the source of learning class-shared or class-specific patterns.
>
> (2) Different architecture design: we put the negative margin (NM) in the lower layers and positive margin (PM) in the higher layer. In the relation-mapping module, based on our findings of the relevance between class relationship and pattern learning, for the NM (PM) branch, we specifically search for smaller negative margins (larger positive margins) for similar classes, encouraging the model to learn more class-shared (class-specific) patterns, which has novelty in the observation, analysis, architecture design, mapping mechanism design, and hyper-parameter search space.
>
> More detailed comparison will be illustrated below.

---

> > ### Author Response · Authors · 2022-08-02
> > **Response to Reviewer 9Up9 (part2)**
> >
> > # 2. Comparison with methods in face recognition
> >
> > We implemented some methods in the face recognition community that might be relevant to our work, including
> >
> > (1) Adaptive-margin-based methods (CurricularFace (Huang et al. 2020), AdaCos (Zhang et al. 2019), ElasticFace (Zhang et al. 2022), AMR-Loss (Zhang et al. 2021)), such as adjusting the margin value according to the intra/inter-class angles. However, these methods always rely more on the angles across the training time than on angles across all training classes, which differs from our class-relation-based mapping mechanism which relies more on angles across all training classes.
> >
> > (2) Relational-margin-based methods (TRAML (Li et al. 2020)), which maps the class relationship to the margin. However, this work takes the semantic embedding (such as attributes) of each class as input, and utilizes a network to learn the relational margin, which differs from our work in that we don’t need further training a network nor the attributes to obtain the relational margin. Moreover, we specifically design the search space of hyper-parameters of the linear mapping to allow similar classes to have a margin with a larger absolute value, so that the mapping is more interpretable than mapping by a learnable black-box neural network.
> >
> > Below we empirically compare our method with the above face recognition methods on CIFAR100 following settings in our paper. Our aim of experiments includes (1) verifying whether they can solve the dilemma between base-class performance and novel-class generalization and (2) verifying whether they can effectively capture the relationship between classes.
> >
> > |CIFAR100 | overall | novel | base |
> > |---|---|---|---|
> > |baseline | 47.02 | 37.4 | 72.32 |
> > |baseline + CurricularFace   			 |	47.22 |	35.77 |	72.67|
> > |baseline + AdaCos	          			|	44.48 |	26.07 |	72.55|
> > |baseline + TRAML 	          			|	47.31 |	36.32 |	73.15|
> > |baseline + ElasticFace  	   			|	47.09 |	36.60   |	72.40|
> > |baseline + AMR-Loss          			|	46.66 |	33.37 |	72.58|
> > |baseline + NM/PM (ours)     			|	**49.21** |	**40.22** |	**73.72**|
> > |baseline + NM/PM + CurricularFace	 |	49.48 |	40.07 |	74.10|
> > |baseline + NM/PM + AdaCos  		  | 	45.24 |	27.20   |	73.83|
> > |baseline + NM/PM + TRAML   		  | 	48.18 |	39.35 |	73.73|
> > |baseline + NM/PM + ElasticFace  		 |	49.43 |	40.70   |	74.09|
> > |baseline + NM/PM + AMR-Loss 		|	49.22 |	38.92 |	74.15|
> > |baseline + NM/PM + relation (ours)	|  	**50.25** |	**41.17** |	**74.20**|
> >
> > From this table, we can see that (1) these methods cannot solve the dilemma by adding directly to the baseline method, since none of them can achieve higher base and novel accuracy simultaneously compared with the baseline performance, while ours (baseline + NM/PM) can; (2) with the NM/PM architecture design, these methods can hardly capture the relationship between classes, since they could not achieve performance significantly higher than the baseline + NM/PM ones, which verifies the effectiveness of our method under the few-shot class-incremental learning task for general data recognition. We add this experiment to the revision (See Appendix section E).
> >
> > # 3. Axis meaning
> >
> > The horizontal axis denotes the adopted margin value, and the vertical axis represents the accuracy (left, L92-93), L1 norm (red, mid, L107-110), activation value (blue, mid, L124-125), other-class activation (red, right, L133), cosine similarity (blue, right, L146-147). We add this illustration to the revision (See Fig.2).
> >
> > # 4. Experiments on face recognition datasets
> >
> > Due to the time limit, we could hardly have enough time to prepare, train, and tune hyper-parameters for the face recognition datasets from scratch. We would further tune hyper-parameters before the final version and add this experiment to the final version.
> >
> >
> > # Reference
> >
> > Huang, Yuge, et al. "Curricularface: adaptive curriculum learning loss for deep face recognition." proceedings of the IEEE/CVF conference on computer vision and pattern recognition. 2020.
> >
> > Zhang, Xiao, et al. "Adacos: Adaptively scaling cosine logits for effectively learning deep face representations." Proceedings of the IEEE/CVF Conference on Computer Vision and Pattern Recognition. 2019.
> >
> > Boutros, Fadi, et al. "Elasticface: Elastic margin loss for deep face recognition." Proceedings of the IEEE/CVF Conference on Computer Vision and Pattern Recognition. 2022.
> >
> > Zhang, Zhemin, Xun Gong, and Junzhou Chen. "Face recognition based on adaptive margin and diversity regularization constraints." IET Image Processing 15.5 (2021): 1105-1114.
> >
> > Li, Aoxue, et al. "Boosting few-shot learning with adaptive margin loss." Proceedings of the IEEE/CVF conference on computer vision and pattern recognition. 2020.

---

> > > ### Author Response · Authors · 2022-08-08
> > > **Response to Reviewer 9Up9 (Face datasets experiments)**
> > >
> > > Thank you very much for reading our response. May I know if our response have addressed your questions? We would like to report our method implemented on face datasets as follows, where we follow (Wen et al. 2022) to train on VGGFace2 and test on other datasets.
> > >
> > > |       |LFW|AgeDB-30|CA-LFW|CP-LFW|Combination|
> > > |-----|------|------|-------|-------|------|
> > > |baseline|99.46|92.56|92.88|90.90|93.67|
> > > | + Ours|99.60|94.03|93.36|91.35|94.39|
> > >
> > > From these results, we can see improvements compared with the baseline method.
> > > Please feel free to let us know if you have any question. We are very much looking forward to having the opportunity to discuss with you!
> > >
> > > ## Reference
> > > Wen, Yandong, et al. "SphereFace2: Binary Classification is All You Need for Deep Face Recognition." International Conference on Learning Representations. 2022.

---

> ### Author Response · Authors · 2022-08-09
> **Looking forward to having the opportunity to discuss with you**
>
> Thank you very much for reading our response! May I know if our response have addressed your questions? Please feel free to let us know if you have any question. We are very much looking forward to having the opportunity to discuss with you.

---

### Official Review · Reviewer_L5yy · 2022-07-10

**Rating:** 6
**Confidence:** 5
**Soundness:** 3 good
**Presentation:** 3 good
**Contribution:** 3 good

**Summary:**

This paper studies margin-based few-shot class-incremental learning (FSCIL), which is motivated by an observation that positive (negative) margins adopted in base training can negatively influence the transferability (discriminability) of the representations. The authors explained this dilemma as a class-level overfitting problem from the perspective of pattern learning and claimed the cause of such class-level overfitting is due to the fact that the constraint of learning shared or class-specific patterns can be easily satisfied. To address this problem, the authors proposed simultaneously adopt positive and negative margin and learn positive-margin-features from negative-margin features. Experiments on a set of FSCIL datasets show the proposal can achieve STOA performance.

**Questions:**

1. How to justify the claim about the dilemma is from the easily-satisfied constraint of learning shared (e.g., corner or edge features) or class-specific patterns (complex class-specific features that cannot generalise).
2. How $m_{upper}$ and $m_{ave}$ are set?
3. Why baseline method performs so well?
4. the authors find large margin can lead to sparse patterns. Is this observation to some extent relevant to supervision collapse observed in [1], which described lose any information that is not necessary for performing the training task, including information that may be necessary for transfer to new tasks or domains?

**Limitations:**

Yes

**Strengths And Weaknesses:**

Paper Strength

+ The connection between margins and pattern learning is interesting. It is an interesting observation that positive margin pushes the sparsity and fitness of patterns but harms the transferability of the patterns. This observation provides insight in understanding the generalisation of deep neural networks.
+ Thorough experiments are conducted to evaluate the effectiveness of the proposed method. The results reveal that the method can simultaneously improve the performance of base and novel classes and thus is empirically proven to be able to overcome the dilemma of the margin-based classification.

Weakness
- There lacks evidence to support some important claims made in this paper. For example, the authors claimed the cause of the class-level overfitting problem lies in the easily-satisfied constraint of learning shared (e.g., corner or edge features) or class-specific patterns (complex class-specific features that cannot generalise). But how to verify this? Additional visualization seems necessary.
- It makes sense to consider the class relations into class-specific margin. But the necessity to setting two key hyper-parameters $m_{upper}$ and $m_{ave}$ manually makes the method less elegantly. How these two parameters are determined? Is this method sensitive to such parameters?
- From Table 3 it can be the baseline method performs very well even better than most of the comparing methods. More detailed explanation to the baseline and its performance should be added.

---

> ### Author Response · Authors · 2022-08-02
> **Response to Reviewer L5yy**
>
> We thank the reviewer for the valuable advice. We address each below.
>
> # 1. Verification of easily-satisfied constraints
>
> We would like to first clarify that the easily-satisfied constraint refers to easily learning simple patterns shared between classes given negative margins, and we just use corners or edges for an example. Also, given positive margins, the easily-satisfied constraint refers to easily learning complex patterns only specific to a certain class, and the complex texture is just an example.
>
> For justifying that the generation of the dilemma is the result of the easily-satisfied constraint, we need to analyze how simple or how complex the learned patterns are. However, qualitative experiments like visualization of patterns are subjective to some extent, and due to the small input image size (e.g., 32 $\times$ 32 for CIFAR100), small visual areas such as textures are hard to see. Instead of qualitative verification by visualization, we are inspired by (Kornblith et al. 2019) to quantitatively measure the simplicity/complexity of patterns by the similarity between the extracted feature and the simplest feature (e.g., corner or edge features).
>
> We first use the baseline model to train on CIFAR100, and use the first convolution layer as the simplest feature extractor (denoted as $f_{simple}$), since many works (e.g., (Yosinski et al. 2015)) has shown that the first convolution layer tends to capture corners or edges. Then, we train models with and without our proposed methods with different margins, and use the backbone network for feature extraction (denoted as $f_{target}$). After that, we extract $f_{simple}$ and $f_{target}$ features from all images in base classes. Finally, we compare the CKA similarity (Kornblith et al. 2019) for measuring the similarity between $f_{simple}$ and the $f_{target}$.
>
> For a sanity check, we first report the similarity between different layers within the baseline model.
>
> | $f_{target}$         |  CKA                |
> | ---------------          | ------------------- |
> | Conv1-output      |  1.0   	       |
> | Stage1-output     |  0.8876           |
> |Stage2-output      |  0.5664           |
> |second-last-Conv |  0.2097           |
> backbone-output   |  0.1306           |
>
> We can see that the shallower the layer is, the higher the CKA similarity would be, which means the more similar they are to the $f_{simple}$, i.e. the simpler the pattern would be (Yosinski et al. 2015), and the more transferable but less discriminative the feature would be [25]. Then, we report the comparison of the CKA similarity between $f_{simple}$ and $f_{target}$ (backbone feature) of the baseline model trained with margins.
>
> |Margin    |-0.5       |-0.4         |-0.3        |-0.2        |-0.1        |0.0          |0.1         |0.2         |0.3         |0.4          |
> |------------|------------|------------|------------|------------|------------|------------|------------|------------|------------|------------|
> |CKA 	|0.2432   |0.2245   |0.2010   |0.1661   |0.1510   |0.1306   |0.1149   |0.0837   |0.0642   |0.0576      |
>
> We can see that by applying a negative margin, the CKA similarity clearly increases (even larger than that of the second last convolution layer when margin < -0.3), which means the backbone network captures patterns more similar to simplest ones such as edges or corners. And **this is the verification that the model tends to learn simple patterns that are easily shared between classes**. When applying a positive margin, the captured patterns grow to be more complex and tend to overfit base classes, making the CKA much smaller than the baseline, **which is the verification that the model tends to learn complex patterns that are easily to be specific to a given base class**.

---

> > ### Author Response · Authors · 2022-08-02
> > **Response to Reviewer L5yy (part2)**
> >
> > By applying our method, for the negative-margin-based (NM) patterns, we set $f_{target}$ to be the output of the backbone network (i.e., $f(\cdot)$ in Fig.5) and set the positive margin to 0.3, and report the CKA below.
> >
> > |     Margin    |     -0.5    |     -0.4     |     -0.3     |     -0.2     |     -0.1     |     0.0     |
> > | ----------------|------------|------------|------------|------------|------------|------------|
> > |   CKA (baseline) |0.2432 |  0.2245  | 0.2010 |  0.1661 |  0.1510 |  0.1306 |
> > |   CKA (NM) |   0.1867 |  0.1779  |  0.1724  |  0.1638  |  0.1552  |  0.1427  |
> >
> > We can see the CKA similarity decreases clearly compared with the baseline method with small margins, which validates that the negative-margin-based patterns learned by our method are more complex and less similar to edges and corners than the baseline method, **verifying the mitigation of the easily-satisfied constraint problem by extra supervision from the learning of positive-margin-based patterns**.
> >
> > Also, we report the CKA for the positive-margin-based (PM) patterns by setting the $f_{target}$ to the output of the $F(\cdot)$ in Fig.5, and set the negative margin to -0.4 below.
> >
> > |Margin  | 0.0 |  0.1 | 0.2 | 0.3 | 0.4 |
> > | ---  | ---  | ---  | ---  | ---  | ---  |
> > |CKA (baseline) | 0.1306  | 0.1149  | 0.0837 |  0.0642 |  0.0576 |
> > |CKA (PM) |0.1476 | 0.1439 | 0.1430 | 0.1214 | 0.1100 |
> >
> > We can see that the CKA is larger than that of the baseline method with large margins, which validates that the positive-margin-based patterns learned by our method is more similar to the simplest patterns such as edges or corners than the baseline method, which makes it less overfitting the base classes, **verifying the mitigation of the easily-satisfied constraint problem by extra supervision from the learning of negative-margin-based patterns**.
> >
> > In summary, the above experiments validate that the baseline model tends to learn simple patterns that are easily shared between classes given negative patterns, while it tends to learn complex patterns that are easily overfitting a base class given positive margins. Moreover, by applying our method, such easily-satisfied constraint problems are mitigated. We add this experiment in the revision (See appendix section D).
> >
> > # 2. Determination and sensitivity of $m_{ave}$ and $m_{upper}$
> > We would like to point out that the setting of the hyper-parameters is illustrated in the paper L241-242. Specifically, we first set $m_{ave}$, without applying the relation mapping module, to the ordinary cosine margin applied in Eq. 5, because the average of class relations reflects a global margin that is effective for most classes (L238). Then, we fix $m_{ave}$ and search for the $m_{upper}$. Suppose the search spaces for $m_{ave}$ and $m_{upper}$ are $S_{ave}$ and $S_{upper}$, the complexity of searching for optimal $m_{ave}$ and $m_{upper}$ is O(|$S_{ave}$| + |$S_{upper}$|) rather than O(|$S_{ave}$| * |$S_{upper}$|) of the ordinary grid search, which means the determination of these two hyper-parameters is much easier than the ordinary grid search.
> >
> > We studied the sensitivity of both parameters in Fig. 7, 8 and Fig.9 (sub-figures denoted as All Classes). Specifically, in Fig. 7, we can see that negative $m_{ave}$ for negative-margin-based patterns gives a stable performance improvement when margin is in [-0.4, -0.1]. Similarly, $m_{ave}$ for the positive-margin-based patterns in Fig. 8 shows a stable performance improvement when margin is in range [0.1, 0.3], $m_{upper}$ consistently promotes to the performance when in [-0.6, -0.3] and $m^P_{upper}$ stably improves the performance when in [0.4, 0.8] in Fig. 9. The above non-narrow range of hyper-parameters indicates that the improvements of our method are not sensitive to the two hyper-parameters.
> >
> > # 3. Baseline performance.
> >
> > We would like to point out that our implementation is directly based on the code released by CEC [26] and will be released (L265). For the CIFAR100 and miniImageNet, the baseline performance is identical to previous works.  For CUB200, our baseline overall performance is not higher than [4]. On CUB200, since the pre-training of the backbone is adopted following [28, 26], we scale the learning rate of the backbone network to 10% of the global learning rate (L266-267), which is a well-known method in finetuning (e.g., Stanford cs231: cs231n.github.io/transfer-learning).
> > Note that the improvements of our method are not fully attributed to the baseline performance, our analysis and solution to the dilemma works on CIFAR100 and miniImageNet and gives state-of-the-art performance, which validates the effectiveness of our insight and method.

---

> > > ### Author Response · Authors · 2022-08-02
> > > **Response to Reviewer L5yy (part3)**
> > >
> > > # 4. Relation to supervision collapse
> > >
> > > The class-level overfitting problem in our paper is to some extent relevant to the supervision collapse observed in (Doersch et al. 2020) in that we both observe the patterns learned on base classes tend to represent only the base classes. Moreover, by applying a large margin, this problem is even exacerbated. (Doersch et al. 2020) handles this problem by integrating self-supervised learning and classifying through comparing object parts, so as to reduce the reliance on class labels. Compared with (Doersch et al. 2020), we treat this problem from a different perspective: our work specifically targets the problem caused by applying margins, provides deeper analysis from the aspect of pattern learning, and finally handles this problem by building positive-margin-based patterns from negative-margin-based patterns. In summary, our observations are relevant, but our insight and solutions are different.
> > >
> > > # References
> > >
> > > Kornblith, Simon, et al. "Similarity of neural network representations revisited." International Conference on Machine Learning. PMLR, 2019.
> > >
> > > Yosinski, Jason, et al. "Understanding neural networks through deep visualization." arXiv preprint arXiv:1506.06579 (2015).
> > >
> > > Doersch, Carl, Ankush Gupta, and Andrew Zisserman. "Crosstransformers: spatially-aware few-shot transfer." Advances in Neural Information Processing Systems 33 (2020): 21981-21993.

---

> > ### Comment · Reviewer_L5yy · 2022-08-08
> > **New analysis is encouraged to be added to the paper**
> >
> > I appreciate the explanations and analysis made by the authors, which are encouraged to be added to the paper to justify the claim.

---

> > > ### Author Response · Authors · 2022-08-08
> > > **Thank you for your response!**
> > >
> > > We really appreciate your constructive review and your precious time! We have added the new analysis and experiments to the revision of the appendix (Please see appendix section D), and we would make them concise to add into the paper. If you have any additional concerns please let us know and we would be happy to follow up. Thanks!

---

### Official Review · Reviewer_cyf6 · 2022-07-11

**Rating:** 6
**Confidence:** 3
**Soundness:** 3 good
**Presentation:** 3 good
**Contribution:** 3 good

**Summary:**

This paper tackles Few-shot Class-incremental Learning (FSCIL) problem which is designed to incrementally recognize novel classes with only few training samples after the (pre-)training on base classes. A dilemma exists that we can hardly achieve both good base-class performance and novel-class generalization simultaneously.

This paper analyzes the effects of positive margin and negative margin loss to the novel and based classes and designs a new network in which negative margin loss is attached to the lower layer and positive margin loss is attached to the top layer. Further improvement method based on the adjacency matrix is also proposed. Extensive experiments on three datasets show the effectiveness of the proposed method.

**Questions:**

- For the ablation study, how about the case when the positive margin loss is attached to the lower layer, and negative margin loss is attached to the higher layer? Is the proposed design of the loss function positions really better?

- For the experiment of Sec.4.5, which dataset is used?


**Limitations:**

The authors did not address the potentially negative societal impact of this work.

**Strengths And Weaknesses:**

Strengths
- Enhancing the novel class generality while keeping the discriminative features for the base model with positive and negative margin losses, which are attached to lower and upper layers is an elegant solution.
- Further improvement method based on the adjacency matrix of all classes is proposed.
- Detailed analysis of the dilemma of FSCIL is shown.
- Experimental results show that the proposed method outperforms SOTA methods on FSCIL.

Weaknesses
-  The proposed method is based on existing findings in the paper on the negative margin matters[15]. (Thought the method and analysis of this paper are more sophisticated.)

---

> ### Author Response · Authors · 2022-08-02
> **Response to Reviewer cyf6**
>
> We thank the reviewer for the constructive comments. We address each below.
>
> # 1. Based on existing findings [15] though much more sophisticated
>
> We would like to emphasize that our work significantly differs from [15] in that
>
> (1) Although inspired by the findings in [15], we further analyze the cause of the dilemma between negative and positive margins from the aspect of pattern learning.
>
> (2) Our analysis and interpretation of the dilemma can be directly utilized to develop methods for handling the dilemma (e.g., building positive-margin-based patterns from negative-margin-based patterns), while [15] did not address this problem.
>
> (3) Based on the observation of the relevance between class relationships and pattern learning, we further mitigate the dilemma by designing to map class relations to margins.
>
> (4) We focus on both the base-class performance and novel-class generalization, while [15] only emphasizes the novel-class generalization.
>
> Moreover, our solution is not sophisticated to implement. Therefore, although inspired from [15], our work still provides remarkable novel insights and contributions to the research community.
>
> # 2. Verify the proposed margin positions
>
> We would like to report the experiments with all possible combinations of margins on CIFAR100 below, where the horizontal axis represents the margin attached to higher layer, and the vertical axis denotes the margin for the lower layer.
>
> | margin | -0.5       | -0.4       | -0.3       | -0.2       | -0.1       | 0.0       | 0.1       | 0.2       | 0.3       | 0.4      |
> |  -------- | ---------- | ---------- | ---------- | ---------- | ---------- | ---------- | --------- | --------- | --------- | --------- |
> |-0.5 	|41.16 	|42.51 	|43.30   	|45.68 	|47.93 	|49.33 	|49.25 	|49.27 	|48.53   |48.01	|
> |-0.4 	|42.24 	|42.37 	|44.37 	|45.96 	|48.17 	|48.93 	|48.94 	|48.76 	|49.01   |47.15	|
> |-0.3 	|43.99 	|44.45 	|45.06 	|45.85 	|47.51 	|48.90   	|49.38 	|49.09 	|48.87   |48.08	|
> |-0.2 	|45.05 	|44.71 	|45.61 	|45.85 	|48.2   	|48.08 	|49.6   	|48.65 	|48.53   |48.08	|
> |-0.1 	|47.50   	|47.11 	|46.71 	|46.69 	|47.99 	|48.87 	|48.92 	|49.21 	|48.55   |47.99	|
> |0.0 		|47.38 	|47.70   	|47.90   	|47.94 	|47.43 	|48.48 	|48.67 	|49.14 	|47.96   |47.87	|
> |0.1  	|47.32 	|48.14 	|48.01 	|47.96 	|47.95 	|48.37 	|47.62 	|48.02 	|48.35   |47.32	|
> |0.2  	|47.44 	|46.66 	|47.70   	|47.05 	|47.27 	|47.21 	|47.4   	|47.55 	|47.64   |46.65	|
> |0.3  	|46.61 	|46.23 	|46.29 	|46.21 	|47.01 	|47.11 	|46.73 	|46.37 	|46.55   |45.68	|
> |0.4  	|45.37 	|45.35 	|45.81 	|45.70   	|46.26 	|46.69 	|45.79 	|46.51 	|46.08   |45.56	|
>
> We can see that the margins adopted in the paper (i.e., positive margin on the higher layer + negative margin on the lower layer, the top right area) show clear improvements over the baseline (i.e., margins on both layers are 0). Instead, other combinations of margins (e.g., negative margin on the higher layer + positive margin on the lower layer, the left bottom area) show a much lower performance compared with the baseline, which verifies our choice of hyper-parameters and our insight: build positive-margin-based patterns from negative-margin-based patterns. We add this experiment in the revision (See the appendix secion C).
>
> # 3. Dataset for Sec. 4.5
>
> The dataset for Fig.7, 8, and 9 ($m_{upper}$) is CIFAR100, while that for Fig.9 ($m^P_{upper}$) is CUB200. We add this illustration in the revision (See Fig. 7,8,9 captions).

---

> ### Author Response · Authors · 2022-08-09
> **Looking forward to having the opportunity to discuss with you**
>
> Thank you very much for reading our response! May I know if our response have addressed your questions? Please feel free to let us know if you have any question. We are very much looking forward to having the opportunity to discuss with you.

---

### Meta-Review · Area_Chair_ADgm · 2022-08-31

**Recommendation:** Accept
**Confidence:** Certain

**Metareview:**

This work studied the few-shot class-incremental learning in the margin-based classification. It presented a deeper analysis about the dilemma between the base-class and novel class performance, from the perspective of positive and negative patterns corresponding to positive and negative margins. Although this dilemma had been observed and analyzed in some previous works, the analysis in this work is deeper and novel. The provided method is inspired by the analysis. Although it is simple, but reasonable and effective verified in experiments. The authors also provide convincing responses to most concerns.

In summary, I think this work is well motivated, well writing. It is a professional work, could be accepted to NeurIPS.

**Award:**

No

---

### Decision · Program_Chairs · 2022-09-14

Accept